# Genome-Wide Identification of the NAC Transcription Factors in *Gossypium hirsutum* and Analysis of Their Responses to *Verticillium wilt*

**DOI:** 10.3390/plants11192661

**Published:** 2022-10-10

**Authors:** Shimei Bai, Qingqing Niu, Yuqing Wu, Kunling Xu, Meng Miao, Jun Mei

**Affiliations:** Plant Genomics & Molecular Improvement of Colored Fiber Lab, School of Life Sciences and Medicine, Zhejiang Sci-Tech University, Hangzhou 310018, China

**Keywords:** genome-wide, NAC transcription factors, *G. hirsutum* L., *Verticillium wilt*, phylogenetic analysis

## Abstract

The NAC transcription factors (NACs) are among the largest plant-specific gene regulators and play essential roles in the transcriptional regulation of both biotic and abiotic stress responses. *Verticillium wilt* of cotton caused by *Verticillium dahliae* (*V. dahliae*) is a destructive soil-borne disease that severely decreases cotton yield and quality. Although *NACs* constitute a large family in upland cotton (*G. hirsutum* L.), there is little systematic investigation of the *NACs’* responsive to *V.* *dahliae* that has been reported. To further explore the key NACs in response to *V. dahliae* resistance and obtain a better comprehension of the molecular basis of the *V. dahliae* stress response in cotton, a genome-wide survey was performed in this study. To investigate the roles of *GhNACs* under *V.* *dahliae* induction in upland cotton, mRNA libraries were constructed from mocked and infected roots of upland cotton cultivars with the *V. dahliae*-sensitive cultivar “Jimian 11” (J11) and *V. dahliae*-tolerant cultivar “Zhongzhimian 2” (Z2). A total of 271 *GhNACs* were identified. Genome analysis showed *GhNACs* phylogenetically classified into 12 subfamilies and distributed across 26 chromosomes and 20 scaffolds. A comparative transcriptome analysis revealed 54 *GhNACs* were differentially expressed under *V.* *dahliae* stress, suggesting a potential role of these *GhNACs* in disease response. Additionally, one *NAC090* homolog, *GhNAC204*, could be a positive regulator of cotton resistance to *V. dahliae* infection. These results give insight into the GhNAC gene family, identify GhNAC*s’* responsiveness to *V.* *dahliae* infection, and provide potential molecular targets for future studies for improving *V. dahliae* resistance in cotton.

## 1. Introduction

Unlike animals, which can actively escape under adverse conditions, plants have evolved high-efficiency sensing and signal transduction mechanisms to defend against multiple exogenous biotic and abiotic threats because of being sessile to their surroundings [1]. Biotic stressors, especially bacteria, fungi, and viruses, can cause diseases and trigger the alteration of gene expression and cellular metabolism to change the growth and development of plants [1,2]. In most cases, transcriptional reprogramming is involved for most plants in response to pathogens. Notably, plants have constructed sophisticated gene expression networks that regulate multiple specific pathogen-responsive genes in a coordinated manner, which requires synergy between the different types of transcription factors (TFs) [1]. In addition, by directly binding to the promoter region of different pathogen-related genes, TFs lead to the induced or repressed expression of these genes to improve the plant’s tolerance to biotic stress [2,3]. 

Among the 58 identified families of TFs in higher plants, NACs, which are derived from three known proteins that contain a highly conserved domain in their N-terminal region NAM (no apical meristem), ATAF (*Arabidopsis* transcription activation factor), and CUC (cup-shaped cotyledon), are one of the largest families of transcriptional regulators in plants [2,3]. Genetic and molecular studies have indicated that NAC functions are associated with many biological processes in plants, such as development, hormone signaling, leaf senescence, and secondary cell wall formation [2,3]. In recent years, NACs have been demonstrated to participate in plant defense responses against pathogens, as positive or negative regulators of downstream defense-related genes [3]. In rice, *NAC60* enhances the defense response to fungus by increasing reactive oxygen species’ (ROS) accumulation, callous deposition, and the upregulation of defense-related genes [4]. Similarly, the pathogen-inducible rice NAC gene *NAC096* contributes to immunity against bacterial and fungal diseases by directly binding to the promoters of defense-related genes [5]. *SlNAP1*, a NAC gene, strongly induced by various stresses in tomato, showed a significantly enhanced defense against leaf speck disease and root-borne bacterial wilt disease by directly activating the transcription of multiple genes involved in salicylic acid (SA) biosynthesis [6]. In contrast, *TaNAC30* negatively regulates the resistance of wheat to stripe rust fungus *Pseudomonas syringae pv. tomato* (*Pst*), and the silencing of *TaNAC30* enhanced resistance to *Pst* by inducing a significant increase in the accumulation of H_2_O_2_ [7].

Cotton, the most abundant natural textile material, is an economic crop worldwide that provides more than 50% of the fiber source in the textile industry [8]. However, cotton, especially upland cotton (*Gossypium hirsutum* L.), which is the main species cultivated on a large scale, is strongly susceptible to various biotic and abiotic stressors [9]. *Verticillium wilt* caused by *V. dahliae* has become the largest biotic limitation to cotton production and is a destructive fungal disease affecting cotton yield and fiber quality, currently threatening the global natural textile material supply [10]. Therefore, core resistance genes and regulators should be identified and the mechanisms mediating disease resistance should be thoroughly characterized to develop *V. dahliae*-resistant cotton cultivars. Recently, a NAC gene, *GbNAC1,* identified from *G. barbadense* can significantly enhance resistance to *V. dahlia,* and the regulatory network model of GhmiR164–*GhNAC100* interactions enhanced resistance to *V. dahliae*, implying that NACs might play pivotal roles in *V. dahliae* stress resistance [11,12]. However, a systematic analysis of NACs’ responses to *V. dahliae* in *Gossypium hirsutum* L. has not been completed. In this study, a genome-wide survey was performed to identify 271 *GhNAC* genes in the *Gossypium hirsutum* L. genome. *GhNACs* were classified into 12 groups based on their sequence similarity. In addition, to identify *GhNAC* candidate genes associated with *V. dahliae* resistance, mRNA libraries were constructed from the upland cotton cultivars with the *V. dahliae*-sensitive cultivar “Jimian 11” (J11) and *V. dahliae*-tolerant cultivar “Zhongzhimian 2” (Z2) inoculated with *V. dahliae*. Subsequently, gene expression profiling demonstrated that selected *GhNACs* might be involved in phytohormone and endoplasmic reticulum (ER) stress regulation. Moreover, we identified one *NAC090* homolog, *GhNAC204*, which is induced by *V. dahliae*. The knockdown of *GhNAC204* expression weakened *V. dahliae* resistance, indicating that GhNAC204 could be a positive regulator of cotton resistance to *V. dahliae* infection. This study provides a foundation for an in-depth functional analysis of novel *GhNACs*, which may be useful for improving *V. dahliae* resistance in cotton.

## 2. Results

### 2.1. Identification and Phylogenetic Analysis of GhNACs in Gossypium hirsutum L.

Using the HMM profile of the NAC domain (PF02365) and the NACs’ sequences from *Arabidopsis* (*n* = 33), 271 *NACs* were identified from the reference genome (NAU, v1.1, downloaded from CottonGen) with the names *GhNAC001* to *GhNAC271*. The lengths of *GhNACs* ranged from 156 (*GhNAC028*) to 860 (*GhNAC118*) amino acids, with molecular weights ranging from 17.246 to 94.324 kDa and pI from 4.325 to 10.284 (Appendix A). The 271 GhNAC proteins, which were classified into 12 groups and named I to XII, were used to construct a neighbor-joining phylogenetic tree (Figure 1, Appendix A). Most subgroups contained at least one *Arabidopsis NAC* gene (*AtNAC*), indicating a close relationship between *GhNAC* and other plants. Among the 12 groups, group III contained the most NAC members (50 *GhNACs*), followed by group I (42 *GhNACs*). Group XII had the lowest number of NAC members, with only two *GhNACs* (*GhNAC052* and *GhNAC182*). In addition, *AtNACs*, such as *ANAC012*, *ANAC030*, *ANAC037,* and *ANAC070* in group IV, played pivotal roles in secondary wall biosynthesis and hormone response, which are important for pathogenic or environmental stress adaptation [3]. A total of 38 *GhNACs* with high similarity to the gene members in group IV were clustered in group IV. These results suggest that *GhNACs* in group IV may have functions similar to those of *AtNACs*, which are potential candidate genes involved in disease resistance.

### 2.2. Chromosomal Locations’ Analysis of the GhNACs

Using the genome sequence of *Gossypium hirsutum* L. as a reference, the 271 identified *NACs* were mapped onto chromosomes or scaffolds by MapChart. The 271 *GhNACs* were distributed across 26 chromosomes and 20 scaffolds, with 126 *GhNACs* distributed on the 13 A genome and 121 *GhNACs* allocated to the 13 D genome; the remaining 24 *GhNACs* were located on 20 scaffolds (Figure 2, Appendix A). Specifically, the number of *GhNACs* distributed on each chromosome was uneven. Chromosome A11 contained the highest number of *GhNACs*, with 16 *GhNACs*, compared with 15 *GhNACs* on chromosome D11 from the D subgenome. In contrast, chromosome D10 contained the lowest number of *GhNACs*, with only five genes (Figure 2, Appendix A). In addition, 13 genes were anchored in 12 A-subgenome scaffolds, and 11 genes were found in 8 D-subgenome scaffolds. In addition, many *GhNACs* were clustered within a short distance, such as the top of A01, A11, D01, D04, and D11 and the bottom of A08, A10, A12, D08, and D12. In contrast, almost all the central chromosomal regions lacked *GhNACs*, including the centromere and pericentromere regions (Figure 2). These results agree with the findings of chromosomal location studies that some genes congregate at certain positions on specific chromosomes [13].

### 2.3. Duplications’ Event Analysis of the GhNACs

To reveal the expansion mechanism of the *GhNAC* gene family, gene duplication analysis including segmental duplication and tandem duplication was performed using BLASTN and the coding sequences (cds) of all *GhNACs*. A total of 207 pairs (246 *GhNACs*) of duplicated *GhNACs* was identified in *Gossypium hirsutum* L. (Figure 3, Appendix A). Among these, 195 pairs (245 *GhNACs*) were segmental duplications with different chromosomal distributions. In addition, 12 pairs (18 *GhNACs*) were tandem duplications, including 4 pairs of tandem duplications (*GhNAC020*/*GhNAC021, GhNAC046*/*GhNAC047, GhNAC184*/*GhNAC186, and GhNAC213*/*GhNAC214*), two triplicate repeats of tandem duplications (*GhNAC68*/*GhNAC69*/*GhNAC70* and *GhNAC263*/*GhNAC264*/*GhNAC265*), and one quadruplicate repeat of tandem duplications (*GhNAC87*/*GhNAC88*/*GhNAC89*/*GhNAC90*), suggesting that they may have arisen from the same duplication event (Figure 3, Appendix A). Most duplication events occurred in 135 pairs of *GhNACs* between the At-subgenome and the Dt-subgenome; 38 pairs occurred between chromosomes and scaffolds, whereas only 19, 11, and 4 duplication gene pairs occurred within the At-subgenome, Dt-subgenome, and scaffolds, respectively (Figure 3, Appendix A). Overall, these genes represent approximately 90.8% (246 of 271) of *GhNACs*, indicating that their origin may be from upland cotton genome duplication events. Furthermore, the number of segment duplications was much greater than the number of tandem duplications; therefore, the former may have been the main contributor to *NACs’* expansion in cotton.

### 2.4. DEGs of GhNACs in Response to V. Dahliae Stress

NAC proteins are plant-specific TFs known for their function in plant defense, acting downstream of many immune response pathways [3]. To identify the NACs involved in the cotton response to *V. dahliae* stress, 12 RNA libraries for three biological replicates constructed from mock and *V. dahliae*-inoculated J11 and Z2 were sequenced. About 18.9–24.6 million clean reads, with a Q30 ranging from 92.1–93.6% and a mapping rate ranging from 73.6–95.2%, were obtained (Appendix A). Genes with a differential expression between susceptible and resistant cottons (cutoff fold change ≥ 2 and *p*-value ≤ 0.05) were defined. 

A total of 39 and 15 *GhNACs* were detected in J11 and Z2 among the DEGs, respectively, including 27 upregulated and 12 downregulated *GhNACs* in J11 and 10 upregulated and 5 downregulated *GhNACs* in Z2 (Figure 4A,B). Furthermore, 10 co-upregulated *GhNACs*, including *GhNAC052*, *GhNAC076*, and *GhNAC081*, were co-upregulated in both J11 and Z2, whereas four co-downregulated *GhNACs*, including *GhNAC011*, *GhNAC136*, *GhNAC228*, and *GhNAC230*, were co-downregulated in both J11 and Z2, suggesting that these co-regulated genes may play a role in *V. dahliae* resistance in upland cotton (Figure 4, Appendix A). In particular, the number of upregulated *GhNACs* was significantly higher than that of downregulated *GhNACs*, indicating that the increased expression of *GhNACs* might play a more crucial role than the decreased expression of *GhNACs* in cotton defense responses to *V. dahliae* stress. Notably, the number of *GhNACs* in J11 was significantly higher than that in Z2, almost three times the number of upregulated and 2.5 times the number of downregulated *GhNACs* in Z2. These results likely reflect the fungal growth in J11 and the massive induction and activation of stress-responsive genes, whereas Z2 could hardly be infected [14]. In addition, to verify the reliability of the RNA sequencing results, the relative expression levels of the selected *GhNACs* were investigated using qRT-PCR. The expression patterns of eight selected *GhNACs* were largely similar to our sequencing data, including five upregulated and three downregulated *GhNACs* (Appendix A). Our present results showed that the expression profiles of those *GhNACs* were sufficiently reliable to be used to investigate *V. dahlia**e*-induced transcriptional changes in cotton.

### 2.5. Gene Structure and Cis-Element of Promoters’ Analysis of DEGs in GhNACs

Research has shown that the exon–intron structural diversity plays a key role in the evolution of gene families [13]. To gain further insight into the diversification of *GhNACs* in response to *V. dahliae* stress, the exon–intron organization of DEGs in *GhNACs* was analyzed. The structural analysis of the DEGs revealed that the number of exons in each gene varied among the different groups (Appendix A). Interestingly, similar gene structures were observed in the same group, with gene clustering, which demonstrated that different combinations of exons and introns might lead to diverse gene functions. The number of exons ranged from two to seven. Most DEGs (22/39, 56.41%) had three exons (Appendix A), although *GhNAC242* contained the least number (only two exons) and *GhNAC052*, *GhNAC182*, and *GhNAC224* contained seven exons, the highest number of all genes (Appendix A). In addition, the 1.5 kb genomic sequences upstream of the transcription start site of these DEGs were regarded as putative promoter regions to identify the presence of cis-elements using the PlantCARE (http://bioinformatics.psb.ugent.be/webtools/plantcare/, accessed on 28 August 2022) tool. In the promoter region of the DEGs, in addition to the typical core cis-acting elements, including TATA and CAAT boxes, there were other types of cis-acting elements, the functions of which included stress response, hormone regulation, cellular development, MYB-binding sites, and MYC-binding sites (Appendix A). The results revealed that the ABA responsive element (ABRE), anaerobiosis responsive element (ARE), ethylene responsive element (ERE), and stress responsive element (STRE) were preferentially identified in the *GhNACs’* promoter regions (Appendix A).

### 2.6. Silencing of V. dahliae-Induced GhNAC Alters Disease Resistance in Cotton

To explore whether the *V. dahliae*-induced *GhNACs* were involved in the plant response to *V. dahliae* stress, one *NAC090* homolog, *GhNAC204*, which is induced by *V. dahliae* in both Z2 and J11, was selected for further study (Appendix A). The transient expression of the GhNAC204-GFP fusion protein was conducted by using agro-infiltration in *N. benthamiana* leaves to determine the cellular localization of GhNAC204. As shown in Figure 5A, GFP that represents GhNAC204 localization was detected in the nucleus (Figure 5A). In addition, to verify the transcriptional activation function of GhNAC204, the pGBKT7 vector was used as a negative control to transform the GhNAC204-pGBKT7 into AH109 competent cells. The positive yeast cells were obtained and then cultured in the corresponding yeast medium. The results showed that the normal expression of *GhNAC204* leads to the activation of a downstream reporter gene, so that the GhNAC204-pGBKT7-positive yeast strains’ normal growth in yeast three-deficient medium SD/-Trp/-His/-Ade confirms that GhNAC204 has transcriptional activation activity and may be a transcriptional activator (Figure 5B). 

Given that *GhNAC204* was significantly induced by *V. dahliae* stress, we examined whether it is possible to weaken *V. dahliae* resistance by the knockdown of the expression level of *GhNAC204*. To verify this conjecture, we used VIGS (virus-induced gene silencing) to knock down the expression level of *GhNAC204* in Z2. The expression level of *GhNAC204* was measured using roots from *TRV:00* and *TRV: GhNAC204* plants approximately 4 weeks after infiltration (Figure 6A). Compared with *TRV:00* plants, *TRV:GhNAC204* plants showed severe disease symptoms including leaf chlorosis and wilting, vascular browning in the stem, and an elevated disease index after inoculation with *V. dahliae* (Figure 6B,C). The fungal recovery assay showed that the number of fungal colonies in the stem segments from *TRV:00* plants was far less than that from *TRV:GhNAC204* plants (Figure 6C). Taken together, these data indicated that GhNAC204 could be a positive regulator in cotton resistance to *V. dahliae* infection.

## 3. Discussion

*Verticillium wilt* of cotton caused by the fungal phytopathogen *V. dahliae* is a destructive vascular disease and severely decreases cotton yield and quality worldwide [9,10]. At present, the most effective way to control *V. dahliae* is the development and application of *V. dahliae*-resistant cultivars [10]. Obviously, the identification of key resistance genes and regulators and a thorough characterization of the mechanisms mediating disease resistance will not only provide new insights into dynamic immune regulation in plants but also be critical for the development of *V. dahliae*-resistant cultivars. The NAC gene family is one of the largest transcription factors in higher plants and has been shown to play an important role in plant defense responses to various pathogens by acting downstream of many immune response pathways [3]. Genome-wide analyses of NAC family members have identified NAC in many species, including 117 NACs in *Arabidopsis*, 151 in rice, 163 in poplar, and 152 in soybean and tobacco [3]. However, systematic analysis of *NACs’* responsive to *V. dahliae* in *Gossypium hirsutum* L. has not been completed, and whether *NACs* could play a role in response to *V. dahliae* in *Gossypium hirsutum* L. remains questionable.

In this study, we performed a genome-wide analysis of the upland cotton *GhNAC* gene family to investigate its potential functions in response to *V. dahliae*. A total of 271 putative *GhNACs* were identified in the upland cotton genome. However, *GhNACs* were unevenly distributed on 26 chromosomes and 20 scaffolds, and a subset of *GhNACs* was clustered at the top or bottom of specific chromosomes (Figure 2). These results are similar to those of other cotton species, such as *G. raimondii, G. arboreum*, and *G. barbadense* [15]. In addition, the results of the gene duplication analysis suggested a mass of gene duplication events in the cotton genome. *GhNACs* clustered into 12 tandem duplication event regions on six chromosomes and one scaffold (Figure 3, Appendix A). Moreover, several tandem-duplicated genes were classified into the same subfamily, implying that they may have originated from recent gene duplication events (Figure 3, Appendix A). Additionally, the results of collinearity analysis indicated that segmental duplication with 195 pairs of *GhNACs* was the main approach for the duplication of the *GhNAC* gene family. These results revealed that tandem and segmental duplications were the major factors leading to the expansion of the *NAC* gene family in the upland cotton genome.

To further explore the key NACs in response to *V. dahliae* resistance and obtain a better understanding of the molecular basis of the *V. dahliae* stress response in cotton, two contrasting upland cotton genotypes (highly resistant and susceptible) were used in global expression profiling to investigate the roles of regulatory NACs in *V. dahliae* induction [14]. A total of 54 *GhNACs* were induced among the DEGs and 68.5% (37/54) were upregulated, indicating a potential positive regulatory role for these upregulated *GhNACs* in cotton defense responses to *V. dahliae* stress (Figure 4, Appendix A). The NTL9 protein, which contains a NAC family DNA-binding domain in the N-terminal region, binds to the promoter of the key synthetic enzyme isochorismate synthase 1 (ICS1) and is essential for inducing flagellin-induced ICS1 expression and other SA synthesis-related genes in guard cells and stomatal immunity [16,17]. In our study, the upregulation of NTL9 homologs *GhNAC052* and *GhNAC182* was observed in both J11 and Z2 (Figure 4, Appendix A). It is likely that *GhNTL9* responds to *V. dahliae* infection by activating the SA. Increasing evidence indicates that the ER plays a role in the plant defense response, known as ER stress-mediated immunity, which halts pathogen infection [18]. NAC062, upregulated by ER stress, relays ER stress signaling from the plasma membrane to the nucleus and plays important roles in regulating the unfolded protein response downstream gene expression for cell survival [19]. Two *NAC062* homologs, *GhNAC081* and *GhNAC101,* were upregulated in both J11 and Z2, indicating that they may contribute to *V. dahliae* resistance in cotton (Figure 4, Appendix A). In addition, the ER stress-mediated immunity regulator *NAC089*, which was previously reported to regulate programmed cell death, positively contributed to host resistance against the pathogens *P. capsici* and *Pst* DC3000 [20,21]. Notably, both the *NAC089* genes (*GhNAC030* and *GhNAC158* in this study) identified in J11 were upregulated (Figure 4A, Appendix A). These results indicate a potential positive regulatory role of ER stress-mediated immunity in cotton defense responses to *V. dahliae* stress. The NAC factor JUB1, which acts as a regulatory component of gibberellin/brassinosteroid metabolism and signaling, was considered to suppress *Pst* DC3000-induced defense responses through the accumulation of DELLA proteins [22,23]. Moreover, among the DEGs, two JUB1 homologs (*GhNAC045* and *GhNAC130*) were downregulated compared with the three JUB1 genes (*GhNAC069*, *GhNAC070*, and *GhNAC264*) upregulated in J11 (Figure 4A, Appendix A). This result revealed that JUB1 homologs are also induced by *V. dahliae* and may have opposing functions in the fight against *V. dahliae* infections. Recently, GhmiR164 targets and leads to the degradation of *GhNAC100,* which acts as a suppressor to bind the CGTA-box of the *GhPR3* promoter, thereby enhancing the resistance to *V. dahliae* [12]. Three upregulated *NAC100* genes, *GhNAC121*, *GhNAC236*, and *GhNAC242,* were found in J11 (Figure 4A, Appendix A), further confirming that *NAC100* might play pivotal roles in *V. dahliae* stress resistance. 

At present, only a few studies have been performed on *NAC090*, and its biological function remains unclear. In this study, we found that one *NAC090* homolog, *GhNAC204*, induced by *V. dahliae* in both Z2 and J11, is localized in the nucleus and has transcriptional activation capabilities, which are typical characteristics of transcription factors (Figure 5, Appendix A). The knockdown of *GhNAC204* expression weakened *V. dahliae* resistance, indicating that GhNAC204 could be a positive regulator of cotton resistance to *V. dahliae* infection (Figure 6). However, how GhNAC204 regulates and participates in the disease resistance pathway to improve the resistance of cotton to *V. dahliae* remains unclear, and further mechanistic analysis is needed. Overall, the expression profiles of *GhNACs* in response to *V. dahliae* infection revealed their potential functions in response to challenge by pathogenic fungi and provide potential molecular targets for future studies for improving *V. dahliae* resistance in cotton.

## 4. Materials and Methods

### 4.1. Plant Materials and Pathogen Treatment

Two *Gossypium hirsutum* L. cultivars, the *V. dahliae*-sensitive cultivar J11 and *V. dahliae*-tolerant cultivar Z2, were used to test *NACs’* response to *V. dahliae* stress [24]. Vd080, a highly pathogenic *V. dahliae* strain, was cultured in liquid Czapek medium at 150 rpm at 25 °C for 5 days. The *V. dahliae* concentration was adjusted to 10^6^ spores/mL by hemocytometer counts and inoculated the cotton roots for 24 h [24]. As a mock-inoculation control, roots were inoculated with water. At last, the roots from six different plants in each treatment were mixed separately, replicated three times, and stored at −80 °C for RNA sequencing. For the fungal recovery assay, the cotton stems above the cotyledons were harvested at 21 days post-inoculation (dpi), sterilized with 75% ethanol for 10 min, and then washed three times with sterile water. The stems were then cut into six pieces, each about 1–2 cm, transferred onto Potato Dextrose Agar (PDA) medium supplemented with kanamycin (50 mg/L), and incubated at 25 °C for 3 days [24].

### 4.2. RNA Preparation and Sequencing

RNA library construction and deep sequencing analyses were performed according to our previous research [25]. Total RNA was isolated by using the RNA reagent (Invitrogen, Carlsbad, CA, USA). The quantity and quality of the isolated total RNA were assessed using a NanoDrop OneC Spectrophotometer (Thermo Scientific™, San Diego, CA, USA). For transcriptome sequencing, the enriched mRNAs were purified from the total RNA with magnetic beads attached to oligo (dT) and the strand-specific libraries were sequenced on Illumina Hiseq2500 platform at Novogene (Beijing, China) with pair-end strategy (2 × 150 bp). 

### 4.3. Identification of Differentially Expressed Genes (DEGs) of GhNACs

Clean data from RNA sequencing were aligned to the reference genome (NAU, v1.1, downloaded from CottonGen) by TopHat (version 2.1.0, download from Hangzhou, China, accessed on 25 August 2022). HTSeq was used to calculate the number of short reads aligned to the characterized gene loci and DESeq2 was then used to identify the DEGs (cutoff fold change ≥ 2 and *p*-value ≤ 0.05).

### 4.4. Identification and Annotation of GhNACs

The Hidden Markov Model (HMM) profile of the NAC domain (PF02365) was retrieved from the Pfam database (http://pfam.xfam.org/, accessed on 26 August 2022) and was used as the query to identify all possible NACs’ sequences with HMMER software (http://hmmer.org, accessed on 27 August 2022) with an E-value cutoff of 1 × 10^−5^. The genomic position, protein length, molecular weight, isoelectric point, and exon–intron structure information were obtained through the CottonFGD (https://cottonfgd.org/, accessed on 27 August 2022). 

### 4.5. Phylogenetic Analysis, Chromosomal Mapping, and Duplications’ Event Analysis

Multiple alignments for all of the available and predicted NACs’ full-length protein sequences were performed using Clustal X (version 2.1, download from Hangzhou, China, accessed on 29 August 2022) with manual adjustment for the alignment of the NACs’ domain. A phylogenetic tree was constructed using the neighbor-joining (NJ) method of MEGA 7.0 with the Poisson correction, pairwise deletion, and 1000 bootstrap replicates. A chromosomal localization map was constructed by using the MapChart (version. 2.32, accessed on 29 August 2022). BLAST was used for the pairwise comparison of the filtered NACs’ sets of *Gossypium hirsutum* L. Gene duplications were analyzed, with the length of the aligned sequence covers being more than 75% of the longer gene and the similarity of the aligned regions being greater than 75% [15].

### 4.6. Expression Analyses of GhNACs

Reverse transcription was performed using the PrimeScript™RT Reagent Kit with gDNA Eraser (Takara, Dalian, China). Quantitative real-time PCR (qRT-PCR) was performed using the TB Green^®^ *Premix Ex Taq*™ (Takara, Dalian, China) on the QuantStudio 3 Applied Biosystem and quantified using the ΔΔC_T_ method. *GhUBQ7* was used as internal controls for mRNA [25]. Primers used for qPCR are listed in Appendix A. The Student’s *t*-test was used for the comparisons between simples. Two-tailed *p*-values of less than 0.05 were considered to be statistically significant [25]. 

### 4.7. Subcellular Localization and Transactivation Activity Assay of GhNAC204

Cells of *A. tumefaciens* GV3101 were transformed with the pGWB405 vector carrying *GhNAC204* under the control of the CaMV 35 S promoter to produce *GhNAC204* with a C-terminal green fluorescent protein (GFP) tag. The transformed GV3101 cells were cultured and used to infiltrate leaves of *Nicotiana benthamiana*. The infiltrated plants were incubated for 48 h; then, green fluorescence was visualized under a confocal laser scanning microscope as previously described [26]. For the assay of the transactivation activity of GhNAC204 protein, the pGBKT7-GhNAC204 recombinant plasmid and the pGBKT7 empty vector were both transformed into the yeast strain AH109 (Takara, Dalian, China). Transformants were grown on SD/-Trp medium for the selection of positive clones and then transferred to SD/-Trp/-His/-Ade medium (Takara, Dalian, China) for the transactivation assay at 30 °C for 3 days.

## 5. Conclusions

In this study, we identified 271 putative *GhNAC* genes in the *Gossypium hirsutum* L. genome, which are distributed across 26 chromosomes and 20 scaffolds and might be derived from polyploidization or segmental duplications. The mRNA libraries of two contrasting upland cotton genotypes (highly resistant and susceptible) were constructed in global expression profiling to investigate the roles of regulatory NACs in *V. dahliae* induction. An analysis of the expression patterns of selected *GhNAC* genes responsive to *V. dahliae* infection was conducted, implying their potential role in resistance to *V. dahliae*. An analysis of the promoter sequences revealed cis-acting elements associated with phytohormone signaling and stress response, with different members harboring distinct types and numbers, suggesting that individual members of the *GhNAC* gene family might be differentially regulated at the transcriptional level. In addition, one *NAC090* homolog, *GhNAC204*, which is induced by *V. dahlia,* was identified. Knockdown of *GhNAC204* expression weakened *V. dahliae* resistance, indicating that GhNAC204 could be a positive regulator of cotton resistance to *V. dahliae* infection. In summary, our study found some key *GhNAC* genes/modules involved in the response to *V. dahliae* stress in upland cotton that provided several candidate targets of genetic modification for further use in resistance to *Verticillium wilt.*

## Figures and Tables

**Figure 1 plants-11-02661-f001:**
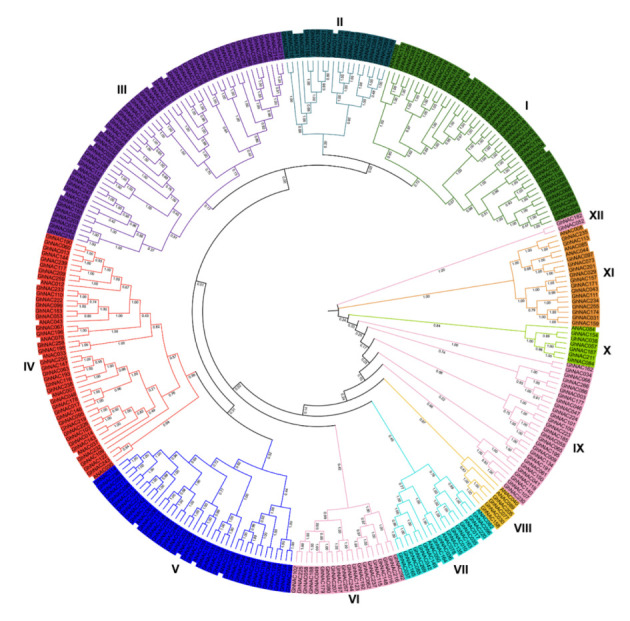
Phylogenetic tree of the 271 GhNAC proteins. Multiple sequence alignment of NAC domain sequences of *G. hirsutum* L. and *Arabidopsis* was performed using Clustal X2. MEGA 7.0 was used to construct the neighbor-joining (NJ) tree with 1000 bootstrap replicates. Various colors indicate different groups of GhNACs.

**Figure 2 plants-11-02661-f002:**
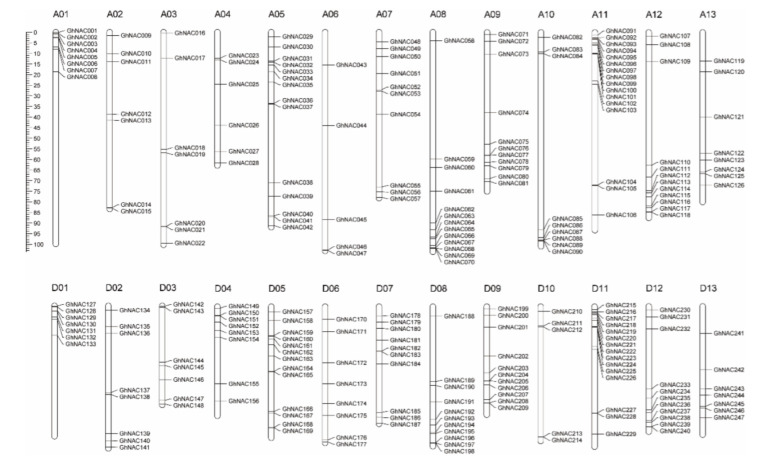
Chromosomal locations of the *GhNACs*. Scale bar on the left indicates the chromosome lengths (Mb). The chromosome numbers of *G. hirsutum* L. (A01–A13, D01–D13) are indicated above each vertical bar.

**Figure 3 plants-11-02661-f003:**
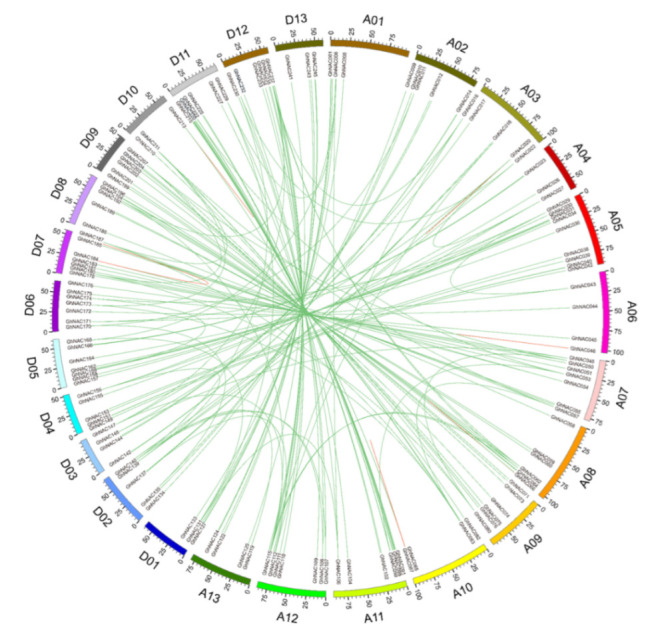
Circos diagram of the *GhNAC* duplication pairs in *G. hirsutum* L. The segmental duplication gene pairs were linked with green lines. The tandem duplicates are denoted by red lines. Scale bar marked on the chromosome indicating chromosome lengths (Mb).

**Figure 4 plants-11-02661-f004:**
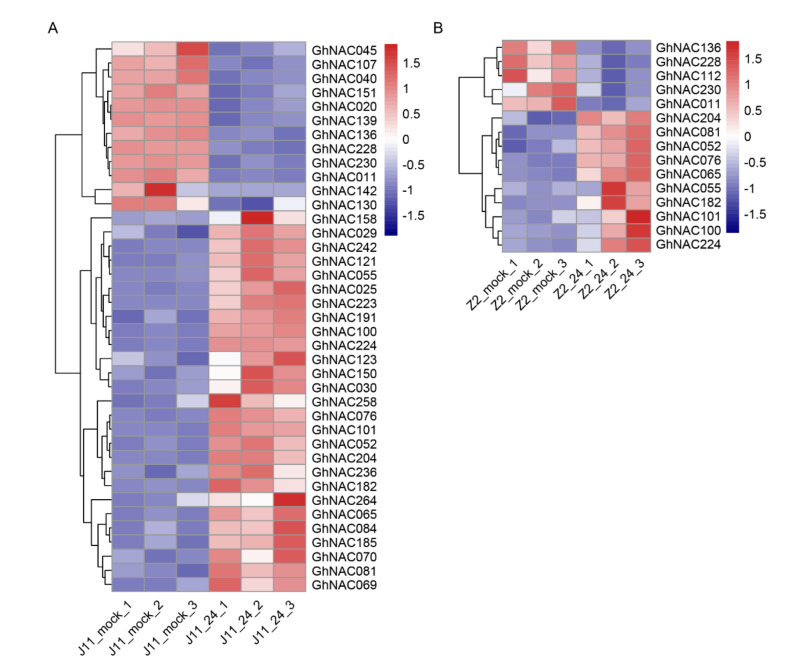
Expression profiling of *GhNACs* in response to *V. dahliae* stress. Hierarchical clustering of differentially expressed *GhNACs* in J11 (**A**) and Z2 (**B**). Red and blue colors show upregulation and downregulation, respectively. The original expression values of the *GhNACs* were normalized using Z-score. The signal intensity ranges from −1.5 to 1.5, as the corresponding color also changes from blue to red.

**Figure 5 plants-11-02661-f005:**
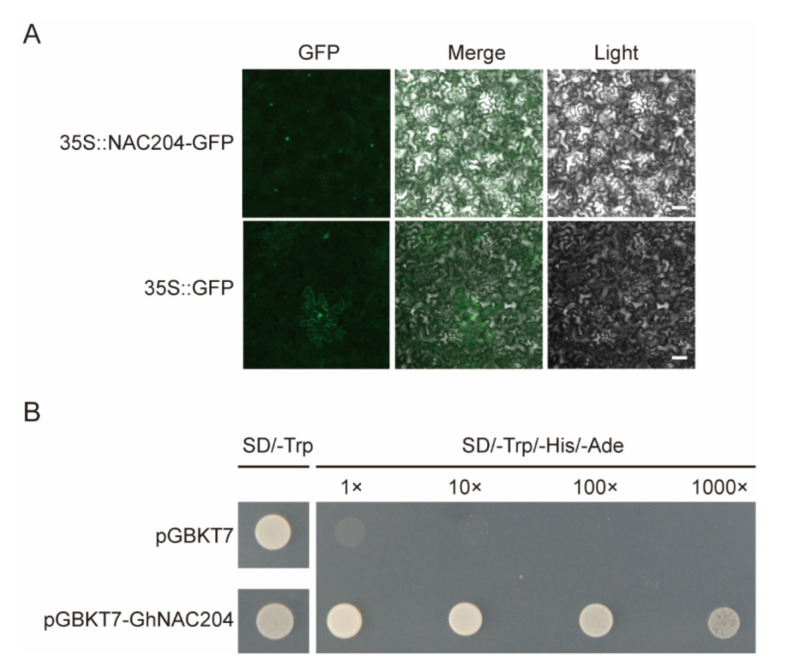
Subcellular localization and transcriptional activity identification of GhNAC204. (**A**) Localization of GhNAC204 in *N. benthamiana* leaves. GhNAC204 was detected on the nuclei; 35 S: GFP as control was detected on the cytoplasm and nuclei; scale bar is 1 mm. (**B**) GhNAC204 has transcriptional activation activity. The GhNAC204-pGBKT7-positive yeast strains’ normal growth in yeast three-deficient medium SD/-Trp/-His/-Ade. SD/-Trp: yeast single deficiency medium; SD/-Trp/-His/Ade: yeast three deficiency medium; pGBKT7: negative control.

**Figure 6 plants-11-02661-f006:**
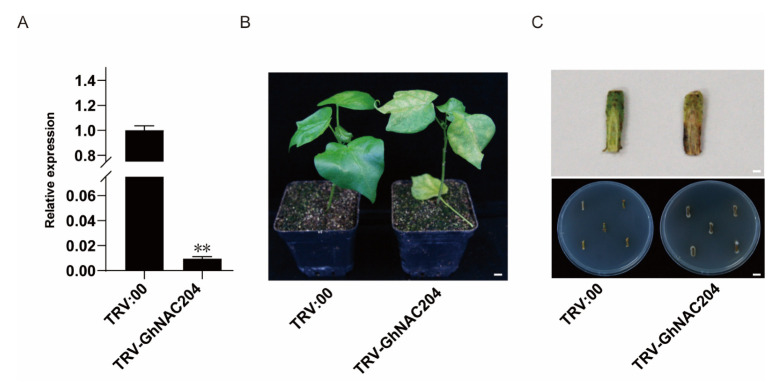
GhNAC204 could be a positive regulator in cotton resistance to *V. dahliae* infection. (**A**) Detection of *GhNAC204* silencing efficiency by qRT-PCR. *GhUBQ7* was used as an internal reference gene and data are means ± SD from three biological replicates; asterisks indicate significant differences when compared with *TRV:00* (*p* < 0.05). (**B**) Silencing of *GhNAC204* leads to severe disease symptoms including leaf chlorosis and wilting; bar = 3.5 cm. (**C**) Silencing of *GhNAC204* leads to more numbers of fungal colonies in stem segments compared with *TRV:00*; upper panel, stem inspection vascular discoloration, bar = 0.3 cm; bottom panel, recovery assay, bar = 2 cm. Photos were taken at 28 days after V. *dahliae* inoculation (40 days after VIGS).

## Data Availability

All the RNA-seq data in this work were deposited in the Sequence Read Archive (SRA) of the NCBI under the accession number PRJNA819185.

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
