# Peer review of "Genome-Wide Identification of the NAC Transcription Factors in Gossypium hirsutum and Analysis of Their Responses to Verticillium wilt"

_plants, 2022, doi:10.3390/plants11192661_

Round 1

Reviewer 1 Report

In the manuscript by Bai et al. entitled "Genome-wide identification and expression analysis of the NAC genes responsive to Verticillium dahliae of G. hirsutum L", the authors represent the NAC genes which represent NAC transcription factors (TFs) responsible for wilting in cotton (G. hirsutum) by involving the soil-borne pathogen Verticillium dahliae and their genome-wide variability and gene expression analysis in both susceptible and tolerant upland varieties.

This research is very interesting, advanced and scientifically sound great having a valuable impact on present-day plant research directions on what is happening throughout the World for identifying regulator genes for disease resistance in respective crop species. As most of the studies on NAC TFs are focusing their roles on abiotic stress tolerance in individual plants, this study is unique to judge the TFs role in biotic stress tolerance, i.e. specially involved in wild disease response.  

So generalizing and preliminary work information on this topic is an important task that the authors have tried to cope with. Now I am describing some specific points to be corrected below:

In the abstract, please don’t mention Background, Methods, Results, and Conclusion headings. Only their descriptions are just enough.

Line 25-27: Please change the fonts as per journal’s format.

Line 117: Please define what is ‘At’ and ‘Dt’ chromosomes. I think using A and D genome are good to understand.

Figure 3: Please use Arabidopsis in this analysis to find any synteny.

Figure 5: Please make A and B upside and downside, not by side-by-side. Because the confocal image becomes very small to visualize the sus-cellular localization.

Line 336-337: Don’t mention twice ‘(Gossypium hirsutum L.)’. Instead of it, please write cotton (Gossypium hirsutum L.) only one time.

Line 338: What is ‘V. dahliae’? Fungus or Bacteria or Virus? Please mention what?

Line 339: Please mention how do you adjust the concentration 106 spores/mL?

Line 344: Please mention the size or weight (g) of the root slices? Although here ‘slices’ are not the appropriate word to mention, rather you can choose ‘pieces.

Line 345: Please mention the full form of PDA.

Line 348, 364-365: Rewrite the line, doesn’t make any sense.

Based on this above-ground of discussion, I would like to author to perform significant revisions in their manuscript as per the given corrections, which will improve it a lot and will be suitable for publication in Plants. 

Reviewer 2 Report

The present manuscript is accompanied by a very well written introduction. An undoubted plus of the presented work is a very good bioinformatic analysis of the results obtained as a result of the analysis of the reference genome - described in chapters:

2.1. Identification and phylogenetic analysis of GhNACs in Gossypium hirsutum L.

2.2. Chromosomal locations analysis of the GhNACs

2.3. Duplications event analysis of the GhNACs

However, I have serious doubts about the description of the experiment in chapter 2.4 and especially about the far-reaching relationships that will be worked out later. Only 8 of the 39 GhNACs identified were experimentally confirmed - far too few to speak of Genome-wide identification and expression analysis of the NAC genes.  More experiments should be performed. Or other parts of the article and its title should be reworded accordingly - it could read: Genome-wide Identification and Partial Expression Analysis of the NAC genes expressed on Verticillium dahliae from G. hirsutum L.

Round 2

Reviewer 1 Report

Now, this manuscript can be accepted for publication.

Reviewer 2 Report

In present form is good.